# Integrated impacts of synoptic forcing and aerosol radiative effect on boundary layer and pollution in the Beijing-Tianjin-Hebei region, China

Yucong Miao[1], Huizheng Che[1], Xiaoye Zhang[1], Shuhua Liu[2]

[1]State Key Laboratory of Severe Weather & Key Laboratory of Atmospheric Chemistry of CMA, Chinese Academy of Meteorological Sciences, Beijing 100081, China
[2]Department of Atmospheric and Oceanic Sciences, School of Physics, Peking University, Beijing 100871, China

Correspondence to: H. Che (chehz@cma.gov.cn) and X. Zhang (xiaoye@cma.gov.cn)

**Abstract.** Rapid urbanization and industrialization have led to deterioration of air quality in the Beijing-Tianjin-Hebei (BTH) region with high loadings of $PM_{2.5}$. The heavy aerosol pollutions frequently occur in winter, closely in relation to the planetary boundary layer (PBL) meteorology. To unravel the physical processes that influence the PBL structure and aerosol pollution in BTH, this study combined long-term observational data analyses, synoptic pattern classification, and meteorology-chemistry coupled simulations. During the winter of 2017 and 2018, Beijing and Tangshan often experienced heavy $PM_{2.5}$ pollution simultaneously, accompanying with strong thermal inversion aloft. These concurrences of pollution in different cities were primarily regulated by the large-scale synoptic condition. Using the principal component analysis with the geopotential height fields at the 850-hPa level during winter, two typical synoptic patterns associated with heavy pollution in BTH were identified. One pattern is characterized by a southeast-to-north pressure gradient across the BTH, and the other is associated with the high-pressure in eastern China. Both synoptic types are featured by warmer air temperature at 1000 m AGL, which could suppress the development of PBL. Under these unfavourable synoptic conditions, the aerosols can modulate the PBL structure through the radiative effect, which was examined using numerical simulations. The aerosol radiative effect can significantly lower the daytime boundary layer height through cooling the surface layer and heating the upper part of PBL, leading to the deterioration of air quality. And this PBL-aerosol feedback is sensitive to the aerosol vertical structure, which would be more effective when the synoptic pattern can distribute more aerosols to the upper PBL.

## 1 Introduction

The Beijing-Tianjin-Hebei (BTH) region is the national capital region of China, and covers an area of ~217,156 $km^2$ in the North China Plain. During the last few decades, the BTH has experienced prosperous economic growth and intensive urban expansion, and becomes one of most developed and populous regions in China. Along with the tremendous development, pollution events with massive $PM_{2.5}$ (airborne particles with aerodynamic diameter less than 2.5 μm) frequently occur in the BTH, due primarily to the high emissions (Cheng et al., 2016; Geng et al., 2017; Zhang et al., 2013).

The fate of emitted pollutants is largely governed by the planetary boundary layer (PBL) (Garratt, 1994; Miao and Liu, 2019; Oke, 2002; Stull, 1988), which is the region of the lower troposphere and strongly influenced by the surface. The PBL acts as changeable coupling agents that modulate momentum, heat, moister, and matter between the surface and free troposphere (Baklanov et al., 2011; Miao et al., 2019a; Stull, 1988). In the vertical dimension, the intensity of thermal buoyancy is controlled by the thermal stratification, and the strength of mechanical turbulence is determined by the surface roughness and

the PBL wind. Together these thermal and mechanical PBL processes determine the vertical dispersion and dilution of pollutants and the air replacing from upper levels (Miao et al., 2019a; Oke, 2002; Stull, 1988). Thereby, the depth of PBL, also known as the boundary layer height (BLH), has been extensively utilized to characterize the atmospheric environmental capacity and the dilution volume of pollutants (Stull, 1988; Seidel et al., 2010; Hu et al., 2014; Miao et al., 2015).

Through observational experiments and numerical simulations, the connections between the PBL characteristics and aerosol

pollution in the BTH have been investigated (e.g., Miao et al., 2019b; Quan et al., 2013; Wang et al., 2018a; Ye et al., 2016; Zhong et al., 2017, 2018). The heavy $PM_{2.5}$ pollution events in BTH typically occur under stagnant situations with shallow PBL (Ye et al., 2016; Zhong et al., 2017 and 2018). On a seasonal basis, the heaviest aerosol pollution in BTH occurs in winter, which is not only ascribed to the seasonal changes in emissions and precipitation, but also the shifts in the BLH (Miao et al., 2015, 2018b). With mountains and seas surrounded (Fig. 1), the PBL process/structure and pollution level in the BTH are

usually impacted by the geographical forcings (Chen et al., 2009; Hu et al., 2014, 2016, Miao et al., 2015, 2016, 2017b). Due to the blocking effects of mountains, the momentum exchanging processes between the PBL and the upper free troposphere could be repressed dynamically (Miao et al., 2018; Wang et al., 2018b). Moreover, the local thermal gradient between the mountains-and-plains or land-and-sea can bring on closed circulation systems, and modify the near-surface winds and thermal inversion intensity, leading to the re-circulation and accumulation of pollutants (Chen et al., 2009; Miao et al., 2015, 2017b,

2019a).

In addition to these local-scale surface factors/processes (e.g., friction, thermally induced wind systems, heat fluxes), the large-scale synoptic pattern (e.g., transient systems, thermal advections) plays a role in supplying the foremost driving for the day-to-day variations of BLH and pollution (Hu et al., 2014; Miao et al., 2019b; Stull, 1988; Zhang et al., 2016). Based on the 850-hPa geopotential height (GH) data from 1980 to 2013, Zhang et al. (2016) elucidated the potential linkages between East Asian

Monsoon, synoptic condition, and air pollution in the North China Plain. They found that the stagnant weather condition with southerly and westerly winds would worsen the air quality in North China Plain, and the occurrence of stagnant condition was relevant to the inter-annual and inter-decadal variability of monsoon. The regional transport of pollutants induced by the large-scale synoptic condition is critical to the air quality (Zhang et al., 2019). Although the previous studies have recognized the importance of synoptic pattern and PBL meteorology for the aerosol pollution in the BTH, most of them focused on the short-

term episodes or a specific city  (e.g., Miao et al., 2019b; Quan et al., 2013; Tie et al., 2015; Wang et al., 2019; Zhong et al., 2017). More investigations are warranted concerning (1) the typical synoptic patterns and (2) their impacts on the PBL characteristics and vertical distribution of aerosols in BTH. These aspects are yet to be clearly known, partly due to the absence of continuous PBL observations. In this study, the link among synoptic condition, PBL structure, and aerosol pollution in the

BTH will be examined using the long-term radiosonde measurements collected in Beijing and Tangshan (Fig. 1b): one close to the mountains and the other adjacent to the Bohai Sea.

On the other hand, during the heavy pollution events, the light-absorbing aerosols can cause the upper layer of PBL to be relatively warm (Ding et al., 2016), and the massive aerosols can intensify the PBL stability through scattering the solar radiations, which can lower the BLH and deteriorate the pollution (Miao et al., 2019a; Quan et al., 2013; Sun et al., 2019; Wang et al., 2019; Zhong et al., 2017, 2018). For instance, the unfavorable PBL meteorology and the feedback of aerosol together, were found to be responsible for ~84% of the explosive growth of $PM_{2.5}$ concentration in Beijing during December 2016 (Zhong et al., 2017). The radiative effect of black carbon on PBL is quite sensitive to the vertical distribution of aerosols, which is also modulated by the synoptic pattern (Wang et al., 2018c). Considering that the large-scale synoptic forcing is the first-order meteorological driving factor for the pollution formation/dissipation, it would be necessary to examine the impacts of aerosol radiative effect on BLH on the basis of synoptic analyses. Thus, this study will objectively classify the synoptic patterns over the BTH during winter from 2017 to 2018, and then evaluate the integrated impacts of aerosol radiative effect on PBL structure under typical synoptic conditions using the meteorology-chemistry coupled simulations. The combination of large-scale synoptic analyses and numerical simulations allows us to understand the complicated meteorology-aerosol interaction in the BTH within an integrated framework.

## 2 Data and Methods

### 2.1 Data and synoptic classification

The aerosol pollution levels in BTH are indicated by the hourly measurements of $PM_{2.5}$ mass concentration from 2017 to 2018 in Beijing and Tangshan (Fig. 1b). For each studied city, there are three $PM_{2.5}$ monitoring sites (illustrated by red crosses in Fig. 1b) carried out by the China National Environmental Monitoring Center (CNEMC). Besides, the radiosonde measurements in Beijing and Tangshan were collected to elucidate the complex associations between the PBL meteorology and aerosol pollution. The sounding stations (illustrated by green triangles in Fig. 1b) are equipped with the L-band radiosonde system (Miao and Liu, 2019), which can provide the vertical profiles of pressure, moisture, air temperature, and wind with a fine resolution (~10 m). The sounding balloons are conventionally launched at 0800 and 2000 Beijing Time (BJT) during a day. In addition, the surface meteorological observations (illustrated by black dots in Fig. 1b) were also obtained.

To unravel the predominant synoptic conditions related to the heavy aerosol pollution in the BTH, the 850-hPa geopotential height (GH) fields were analyzed, which were extracted from the National Centers for Environmental Prediction (NCEP) global Final (FNL) reanalysis. The studied region was centred the BTH, covering an area of 106-126 °E in longitude and 29-49 °N in latitude (Fig. 1a), and this is also the region used in the meteorology-chemical coupled simulations. Using the T-mode principal component analysis (T-PCA) (Huth, 1996; Miao et al., 2017a; Philipp et al., 2014), the dominant synoptic patterns in the BTH were objectively classified. The T-PCA has been widely applied to analyse the regional air pollutions from the synoptic perspective, and demonstrated to be a dependable approach to ravel out the influences of large-scale atmospheric

forcing (e.g., Miao et al., 2017a; Stefan et al., 2010; Zhang et al., 2012). Considering that the heavy PM$_{2.5}$ pollution events primarily occurred during winter (Miao et al., 2018), the daily GH fields in the winter months (January, February, November, December) of 2017 and 2018 were classified in this study. In total, 240 daily GH fields were classified.

## 2.2 Meteorology-chemistry coupled simulations

After identifying the typical polluted synoptic pattern, a typical pollution episode that occurred from 26 to 31 December 2017 was selected and simulated using the Weather Research and Forecasting model coupled with Chemistry (WRF-Chem) (Grell et al., 2005). The model domain centred the BTH and covered the most mainland China with a horizontal resolution of 17.5 km (Fig. 1a). The model top was set to the 10-hPa level, and 33 vertical layers were configured below the top. To resolve the PBL structure, 15 vertical layers were set below 2 km above ground level (AGL). For the simulation of chemistry processes,

the RADM2-MADE/SORGAM chemical mechanism (Ackermann et al.,1998; Schell et al., 2001; Stockwell et al., 1990) were used with the Multi-resolution Emission Inventory for China (MEIC) of 2016, which is the most updated and extensively utilized anthropogenic emission data. The physics parameterization schemes used in this work included the Noah land surface scheme (Chen and Dudhia, 2001), the Mellor-Yamada PBL scheme (Nakanishi and Niino, 2006), the WRF Single-Moment-5-class (WSM5) scheme (Hong et al., 2004), the Betts-Miller-Janjic cumulus scheme (Janjić, 1994), and the updated rapid

radiation scheme considering the aerosol radiative effect (Iacono et al., 2008). The initial and boundary conditions (IBCs) of meteorological parameters were configured using the NCEP-FNL reanalysis, and the IBCs of chemical variables were derived from the global model output (http://www.acom.ucar.edu/wrf-chem/mozart.shtml).

The simulations used abovementioned configurations are referred to as the BASE runs, and numerical experiments that turned off the aerosol radiative option were conducted to evaluate the impacts of aerosol radiative effect. These sensitivity experiments

are regarded as the EXP runs hereunder. According to the common strategy for the Air Quality Model Evaluation International Initiative (AQMEII), the selected pollution episodes were simulated as a sequence of four-day time slices (Forkel et al., 2015), including Slice-1 (2000 BJT 24 December to 2300 BJT 28 December) and Slice-2 (2000 BJT 27 December to 2300 BJT 31 December). The first 24-h simulations of each time slice were considered as the spin-up period, and the chemical initial state of each time slice is adopted from the final state of the previous time slice if available.

**3 Results and Discussion**

## 3.1 Linkages between synoptic condition, thermal stability and PM$_{2.5}$ pollution

The time series of daily PM$_{2.5}$ concentrations in Beijing and Tangshan from 1 November to 31 December in 2017 are shown in Fig. 2a, demonstrating several heavy pollution episodes in the BTH. It is worth noting that Beijing and Tangshan often experienced heavy pollution simultaneously. Comparing with the observed potential temperature (PT) profiles in Beijing and

Tangshan (Fig. 2b-c), it is clear that the quick increase (decrease) of PM$_{2.5}$ concentrations usually accompanied with the warming (cooling) of atmosphere above 1000 m AGL. The warming of upper air also could be observed from the vertical

profiles of temperature, and was often accompanied with high relative humidity within the PBL (Fig. S1). The concurrence of warming aloft and increased $PM_{2.5}$ concentration not only occurred from November to December in 2017, but also in other winter months during 2017 and 2018 (Figs. S2-S4). Given that the distance between Beijing and Tangshan is as long as 200 km, the synchronous change of aerosol concentrations and the concurrence of strong thermal inversion aloft, must be relevant to certain large-scale synoptic patterns (Miao et al., 2018). Therefore, it would be necessary to investigate the $PM_{2.5}$ pollution and its influencing factors from the point of view of synoptic condition.

Based on the 850-hPa daily GH fields in winter from 2017 to 2018, the synoptic conditions were classified using the T-PCA (Fig. 3). There are two dominant synoptic patterns – type 1 and type 2, which account for ~70% of the total. The synoptic type 1 occurs most frequently (39.6%). There is a strong southwest-to-northeast pressure gradient across the BTH, supporting strong northwesterly prevailing winds at the 850-hPa level (Fig. 3a). The average daily $PM_{2.5}$ concentrations in Beijing and Tangshan under type 1 are 34 and 62 $\mu g\ m^{-3}$, respectively (Fig. 4a). Under the synoptic type 2, with a southeast-to-north pressure gradient across the BTH at the 850-hPa level, it is the westerly winds dominated over the BTH (Fig. 3b). The occurrence frequency of type 2 is 30%, ranking second among all the identified synoptic types. The average daily $PM_{2.5}$ concentrations in Beijing and Tangshan under the type 2 are significantly higher than those under type 1 (Fig. 4a), which are 92 and 108 $\mu g\ m^{-3}$, respectively. Except for these two dominant types, the occurrence rate of other five synoptic types is 30.4% in total. Among these five less occurred types, it's worth noting that the synoptic type 4 has the highest average $PM_{2.5}$ concentrations (135 $\mu g\ m^{-3}$ in Beijing and 106 $\mu g\ m^{-3}$ in Tangshan), despite its occurrence frequency is merely 5.0 % (Figs. 3d and 4a). Under synoptic type 4, influencing by a high-pressure in eastern China at the 850-hPa level (Fig. 3d), the southerly prevailing winds can cause regional transport of pollutants to Beijing and Tangshan (Miao et al., 2017a; Zhang et al., 2019). To understand the connection between synoptic pattern and PBL structure, the thermal stabilities between 100 m and 1000 m AGL are compared (Fig. 4b). Stronger thermal stabilities are observed under type 2 and type 4, associated with warmer air temperature at 1000 m AGL (Fig. 4b-c), suppressing the development of PBL (Miao et al., 2017a; Hu et al., 2014). Also, moister air could be observed within the PBL under these two types (Fig. 4d), favouring the formation of secondary inorganic aerosols (Zhong et al., 2017 and 2018). Thus, among all the identified patterns, the synoptic type 2 and type 4 are regarded as the representative polluted pattern. In the next section, a pollution episode associated with type 2 and type 4 will be investigated.

## 3.2 Integrated impacts of synoptic pattern and aerosol radiative effect during the selected episode

To unravel the complicated processes leading to the heavy pollution under synoptic type 2 and type 4, a pollution episode occurred at the end of 2017 was selected and simulated using the WRF-Chem model. Fig. 5 presents the vertical structure of simulated PT in Beijing and Tangshan during the episode. Comparing with the observed PT profiles, the warmings of atmosphere aloft from 27 to 29 December in both Beijing and Tangshan were well simulated, with correlation coefficients of 0.91 (p<0.001) in Beijing and 0.94 (p<0.001) in Tangshan. The changes of wind profile in Beijing and Tangshan were also accurately reproduced, with correlation coefficients greater than 0.64 for both the zonal and meridional winds. In Fig. 6, the simulated near-surface temperature, relative humidity, and $PM_{2.5}$ concentration are validated against the observations.

Although discrepancies exist, the simulated temperature, humidity, and $PM_{2.5}$ all demonstrate rationally good agreement with the observations. Besides, comparing the simulations with aerosol radiative effect to those without, the former presents higher $PM_{2.5}$ concentrations, lower temperatures, and higher humidities, resulting in higher correlation coefficients with observations (Fig. 6). Overall, the good model performances (Figs. 5-6) provide a solid basis to utilize the simulation results to elucidate the physical mechanisms underlying the pollution episode.

Based on the model output, the BLH is estimated as the height where the PT first surpasses the minimum PT below by 1.5 K (Nielsen-Gammon et al., 2008; Seidel et al., 2010). The same BLH derivation method has been widely employed in previous PBL studies (e.g., Hu et al., 2014; Miao and Liu, 2019; Nielsen-Gammon et al., 2008), which can explicitly manifest the influences of thermal stability. Fig. 5 shows the time series of simulated BLH in Beijing and Tangshan. It is clear that the warmings of upper air can suppress the daytime BLH on December 27 under synoptic type 4 and on December 28-29 under

synoptic type 2 (Fig. 7a-b). On December 27, influencing by the southwesterly winds, the warmer air mass could be brought to the BTH (Fig. 7d), enhancing the thermal stability and restraining the growth of PBL (Fig. 8a). The southwesterly prevailing winds can transport the pollutants emitted from upstream plain regions to the BTH and further worsen the air quality (Figs. 8d and 9a). Then, the synoptic condition transitioned to type 2 on December 28-29, and the strong thermal inversion and shallow PBL situation in BTH could last until the outbreak of cold advection on December 30 (Figs. 6, 7 and 8). As shown in Fig. 8a-

c, the average BLH in BTH was suppressed to less than 250 m under synoptic type 4 and type 2 from 27 to 29 December, and then increased to 500 m from 30 to 31 December. As a result, massive aerosols were accumulated in the plains of BTH from 27 to 29 December (Figs. 8e and 9b).

During those heavy polluted days, the suspended aerosols may also modify the PBL structure in BTH to some extent (Gao et al., 2015; Wang et al., 2018c; Miao et al., 2019a; Zhong et al., 2018). As the aerosols reduce the solar radiation reaching the

ground, the development of PBL could be suppressed, particularly during the daytime. As shown in Fig. 9c-d, the aerosol radiative effect can impose significant negative perturbations on the daytime BLH. On average, the daytime BLH in the plains of BTH decreased by 84 m (15 %) on December 27 and 93 m (18 %) on December 28-29, and increased the ground-level $PM_{2.5}$ concentrations by 4.3 and 9.0 μg $m^{-3}$, respectively (Fig. 9e-f). The feedback on $PM_{2.5}$ was more prominent in the regions with higher concentrations, where the ground-level $PM_{2.5}$ concentration could increase by 20 μg $m^{-3}$ during the daytime (Fig.

9e-f). Comparing the induced BLH perturbations on December 27 with those on December 28-29, the decrease of BLH was more significant on December 28-29, which may be caused by the larger amount of aerosols suspended within the PBL on December 28-29 (Fig. 9a-b).

On the other hand, the synoptic condition can also modulate the sensitivity of PBL-aerosol feedback through influencing the vertical distribution of aerosols (Wang et al., 2018c). To elucidate the link among synoptic types and aerosol vertical structures,

we examined the south-to-north cross sections of PT and $PM_{2.5}$ cutting through the most polluted region in BTH (Fig. 10). Influencing by the southerly warm advections under synoptic type 4, the lower troposphere had a stronger thermal stratification on December 27 than on December 28-29 under synoptic type 2 (Figs. 10a-b and 11a), leading to more aerosols in the lower PBL on December 27 (Figs. 10c-d). By contrast, the aerosols can be distributed more evenly in the vertical direction on

December 28-29 under synoptic type 2. Fig. 11b presents the average vertical profiles of $PM_{2.5}$ concentration along the cross section between 38 °N and 39 °N, in which the total amounts of $PM_{2.5}$ were almost the same on December 27 and December 28-29 but distributed distinctly. With more aerosols at the upper levels under synoptic type 2 on December 28-29, the daytime aerosol radiative feedback on the PBL thermal structure was enhanced (Figs. 10e-f and 11c). Since the solar radiation is more intense at the upper levels, the elevated aerosol layer can absorb more solar radiations and strengthen the thermal stratification more effectively (Wang et al., 2018c; Huang et al., 2018). Thus, comparing with synoptic type 4, the type 2 can be more conducive for the aerosol radiative feedback.

## 4 Conclusions

To elucidate the link among synoptic forcing, PBL structure, and aerosol pollution in the BTH, this study combined long-term observational data analyses, synoptic classification, and meteorology-chemistry coupled simulations. On the basis of the wintertime $PM_{2.5}$ measurements and radiosonde data in Beijing and Tangshan from 2017 to 2018, the relationships between PBL structure and aerosol pollution were examined. It was found that both cities often experienced high $PM_{2.5}$ concentrations simultaneously, which typically accompanied with strong thermal inversion aloft. The concurrence of heavy pollution in Beijing and Tangshan were regulated by the large-scale synoptic forcings. Using the T-PCA with the 850-hPa daily GH fields during winter, two typical synoptic patterns relevant to the heavy pollution in Beijing and Tangshan were identified. One is characterized by a southeast-to-north pressure gradient across the BTH at the 850-hPa level, leading to westerly prevailing winds to BTH. The other is associated with the high-pressure in eastern China and southerly prevailing winds to BTH. These two types are both featured by warmer air temperature at 1000 m AGL, which can significantly suppress the development of PBL.

Under these unfavourable synoptic conditions, the aerosols suspended in the atmosphere can modulate the PBL structure. A pollution episode at the end of 2017 associated with these typical synoptic types was simulated using the WRF-Chem by turning on and off the aerosol radiative option. The simulation results indicated that the aerosol radiative effect can significantly lower the daytime BLH through cooling the surface layer and heating the upper part of PBL. Thereupon, more aerosols could be accumulated in the lower portion of PBL. Such a PBL-aerosol feedback is sensitive to the aerosol vertical structure, which would be more effective when the synoptic pattern can distribute more aerosols to the upper PBL. At last, although this study highlights the important roles of multi-scale physical processes in the aerosol pollution in the BTH, the chemical mechanisms/processes also should not be deemphasized.

*Data availability.* The reanalysis data can be downloaded from http://rda.ucar.edu/datasets/ds083.2/. The meteorological data in the BTH are available from the China Meteorological Administration (http://data.cma.cn/), and the $PM_{2.5}$ data can be obtained from the CNEMC (http://www.cnemc.cn/). The model data are available by request (chehz@cma.gov.cn).

*Author contributions.* Development of the ideas and concepts behind this work was performed by all the authors. Model execution, data analysis, and paper preparation were performed by YM and HC with feedback and advice from XZ and SL.

*Competing interests.* The authors declare that they have no conflict of interest.


*Acknowledgements.* This study received financial support from National Natural Science Foundation of China (41705002, 41825011), Beijing Natural Science Foundation (8192054), and Atmospheric Pollution Control of the Prime Minister (DQGG0106). The authors would like to acknowledge the Tsinghua University for the support of emission data.

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

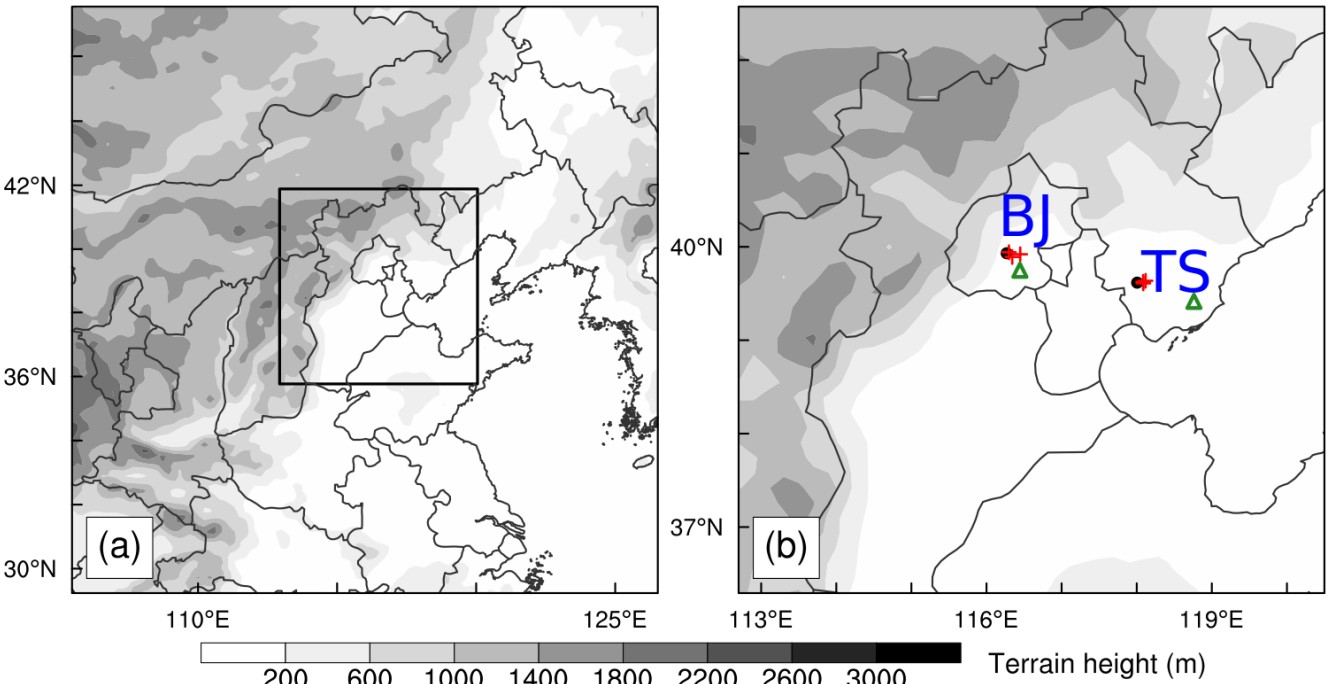

**Figure 1: Maps of terrain height in (a) the simulation domain, and the approximate locations of the Beijing-Tianjin-Hebei (BTH) region is denoted by the black rectangle. In Fig. 1b, the locations of surface meteorological stations and air quality monitoring stations in Beijing (BJ), and Tangshan (TS) are marked by the black dots and the red crosses, respectively. The sounding sites are denoted by the green triangles.**


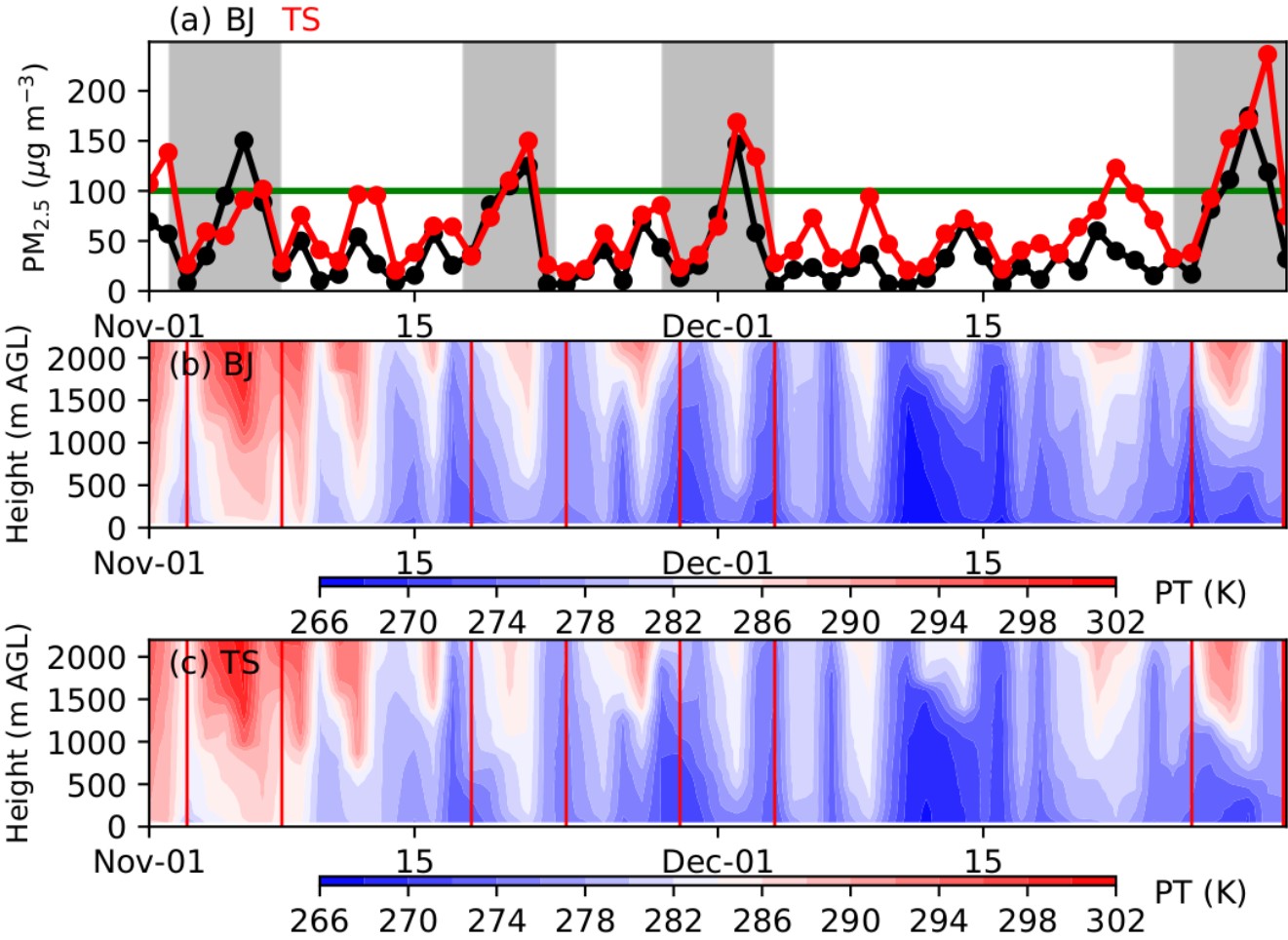

**Figure 2: Time series of observed PM$_{2.5}$ concentration from 1 November to 31 December in 2017 in (a) Beijing and Tangshan, and (b, c) vertical structure of potential temperature (PT) derived from the sounding data at 2000 BJT. Four heavy pollution episodes with maximum daily PM$_{2.5}$ concentration greater than 100 μg m$^{-3}$ in both Beijing and Tangshan are marked by the grey shadings in Fig. 2a.**

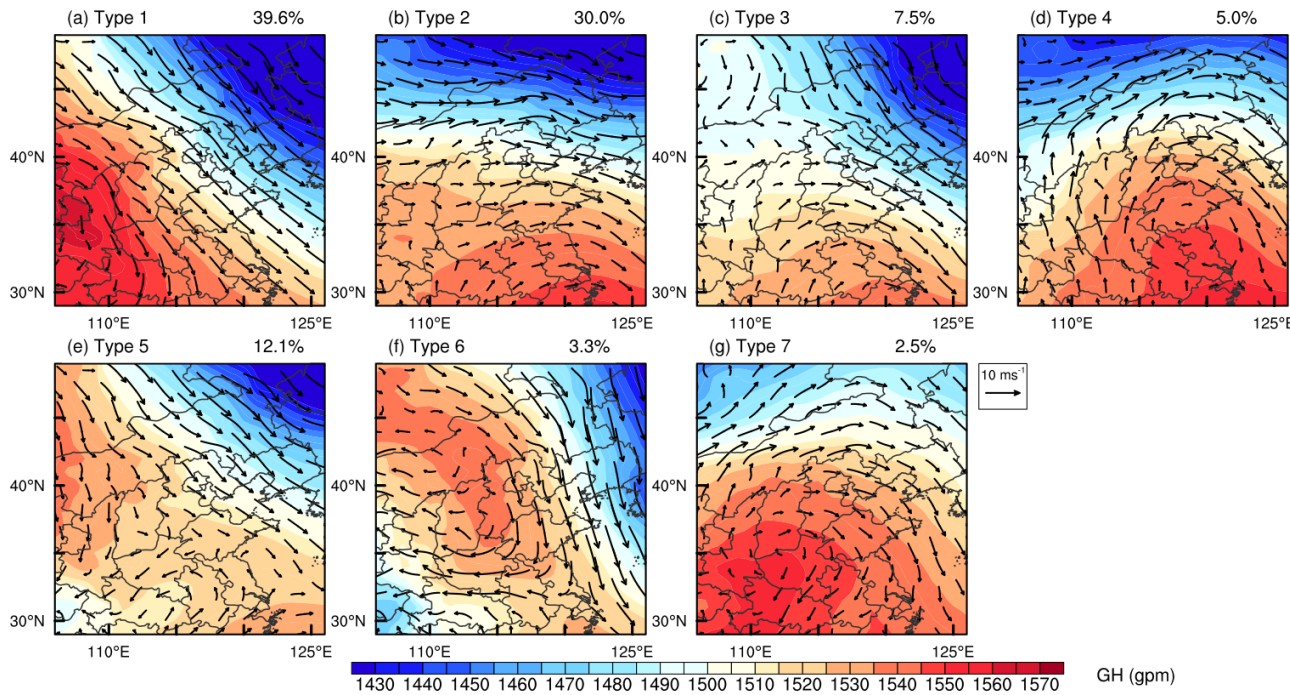

Figure 3: The 850-hPa geopotential height (GH) fields and wind vector fields for the seven classified patterns. The occurrence frequency of each synoptic pattern is also given.

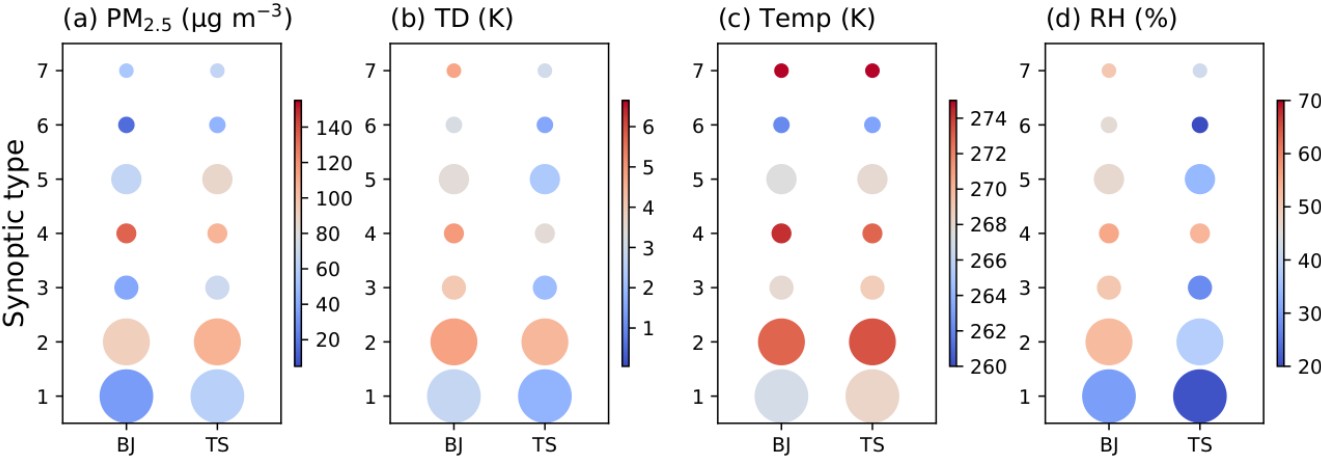

Figure 4: (a) Average PM$_{2.5}$ concentrations under different synoptic conditions in Beijing and Tangshan, and associated (b) thermal differences (TD) of PT between 100 m and 1000 m, and (c) temperature at 1000 m, and (d) relative humidity (RH) at 200 m. The TD equals PT at 1000 m minus PT at 100 m. The size of circle represents the occurrence frequency of each synoptic type. All the meteorological variables shown are derived from the radiosonde data.

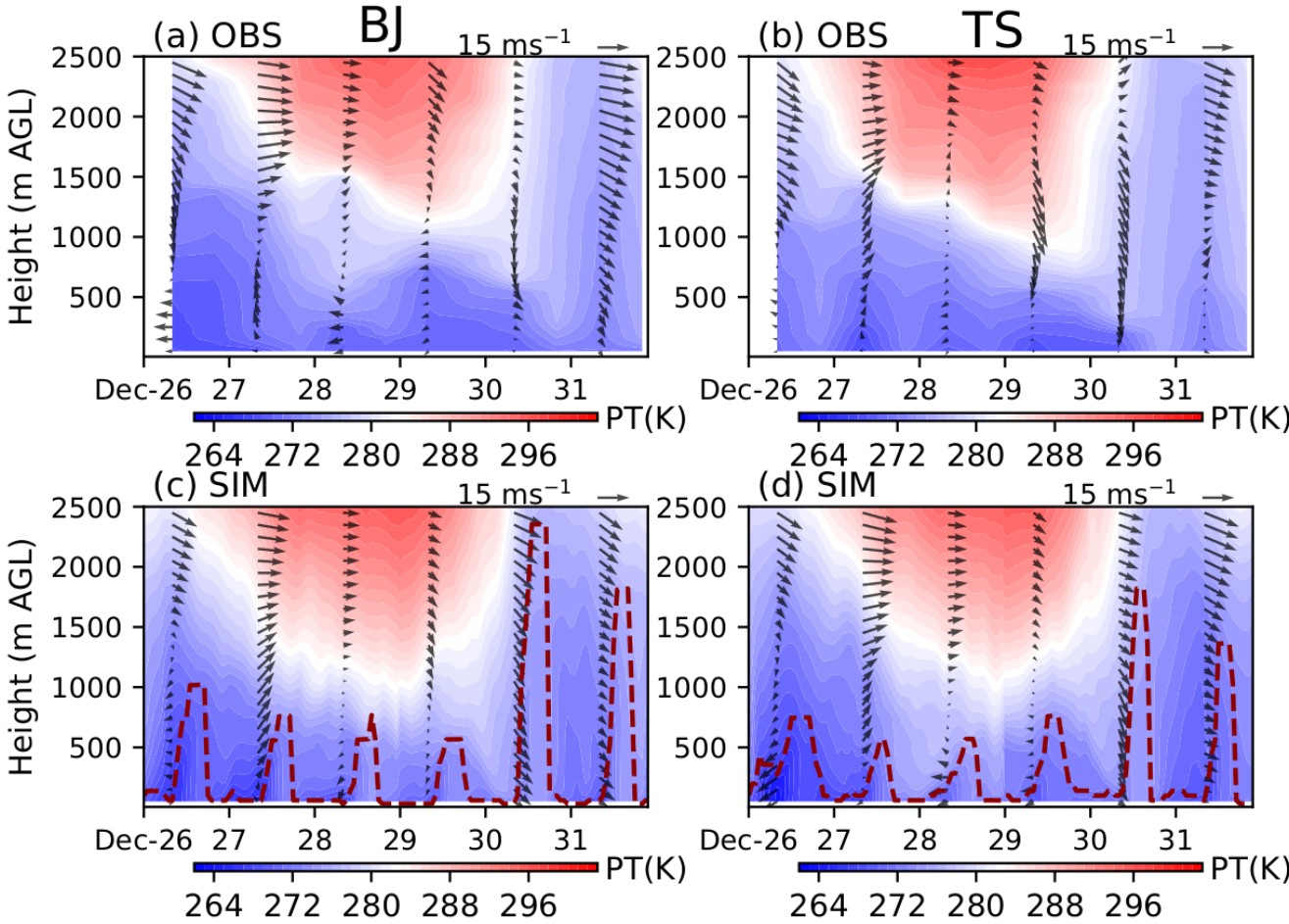

Figure 5: Vertical structure of (a, b) observed and (c, d) simulated PT and horizontal winds in Beijing and Tangshan from 26 to 31 December 2017. The simulated profiles are derived from the BASE run, and the boundary layer height (BLH) is denoted by the red dash lines in Fig. 5c-d.

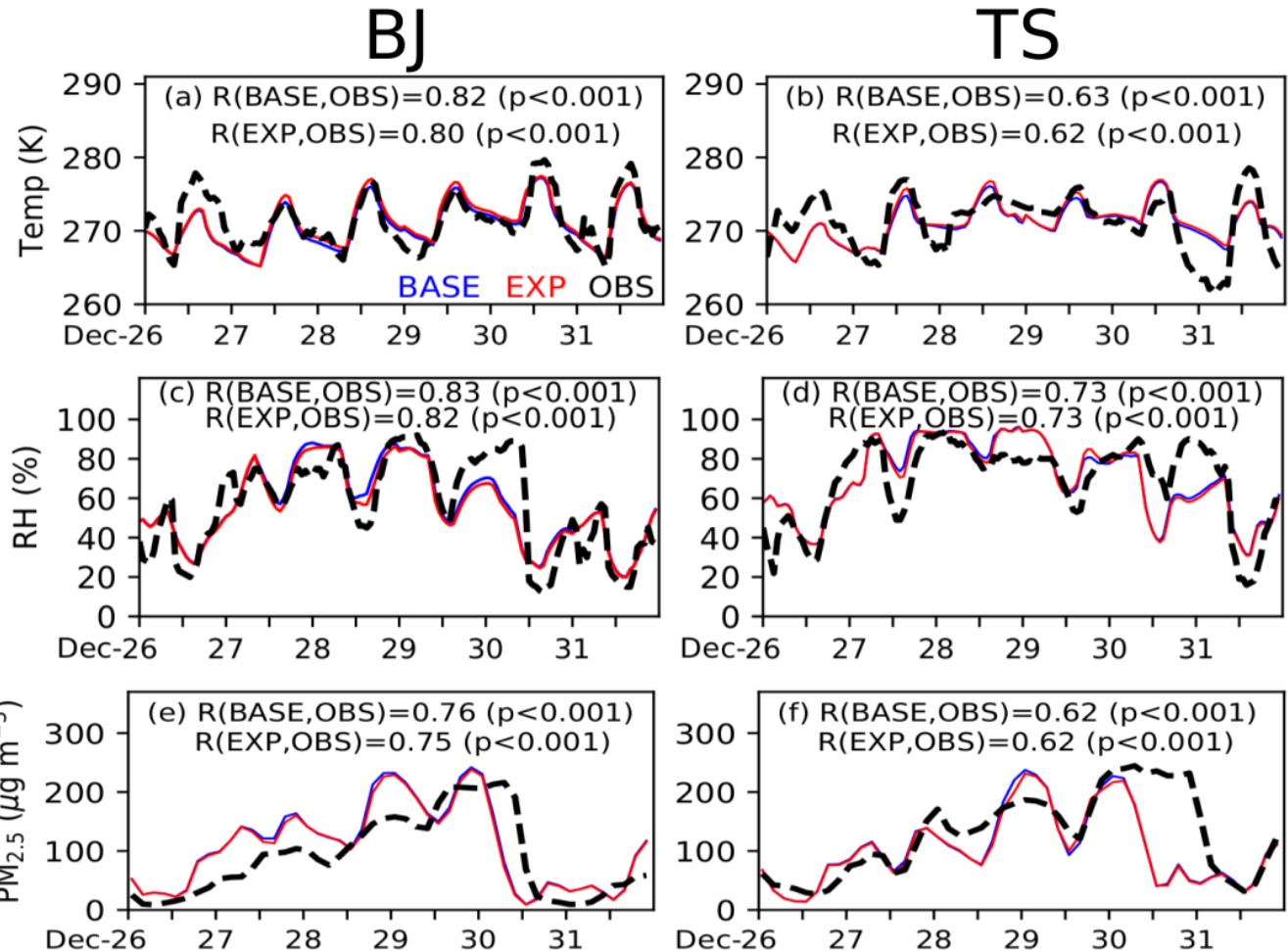

Figure 6: Time series of observed and simulated (a, b) 2 m temperature, (c, d) 2 m RH, and (e, f) PM$_{2.5}$ concentration in (left) Beijing and (right) Tangshan from 26 to 31 December 2017. The simulations of BASE run are denoted in blue lines, and those of EXP run are denoted in red lines. The correlation coefficients (R) between the observations and simulations are also given for each panel.

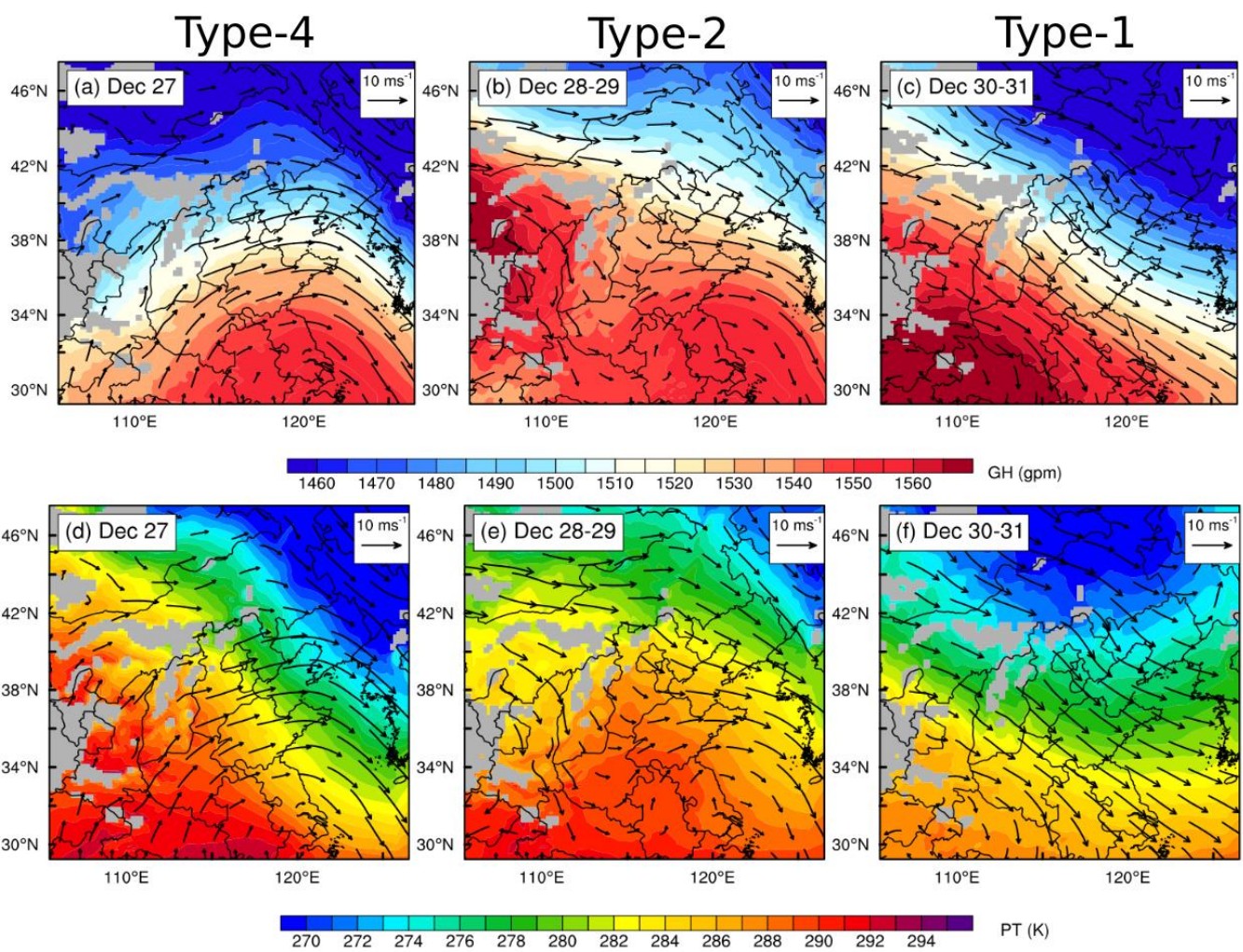

**Figure 7: Simulated 850-hPa (a-c) GH and (d-f) PT fields on December 27, 28-29 and 30-31, overlaid with the wind vectors. The regions with terrains higher than the 850-hPa level are marked by the grey shadings.**

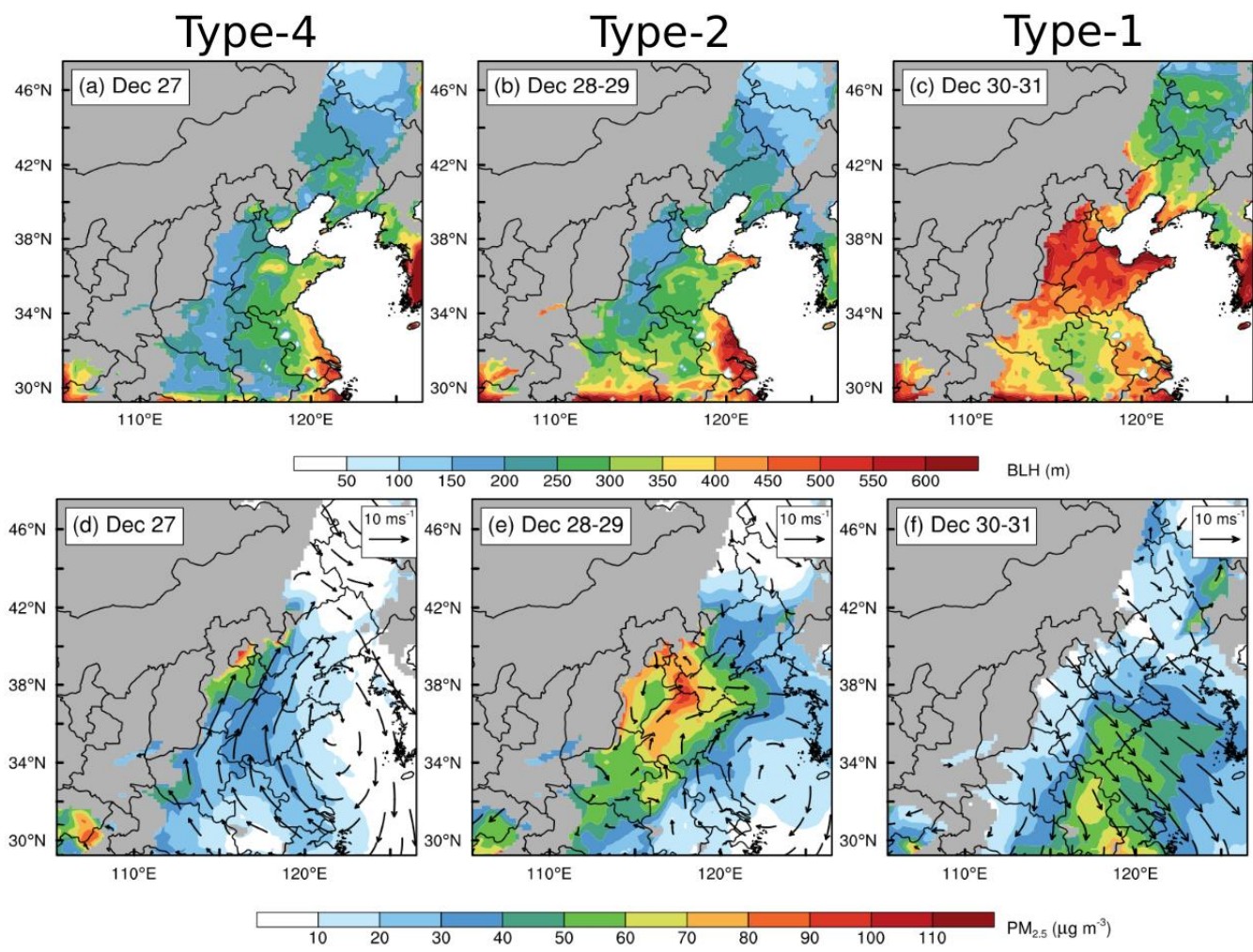

**Figure 8: Spatial patterns of simulated (a-c) BLH and (d-f) 900-hPa PM₂.₅ concentration and wind vectors in the plains of BTH from 27 to 31 December, 2017. The mountainous regions are denoted by the grey shadings.**

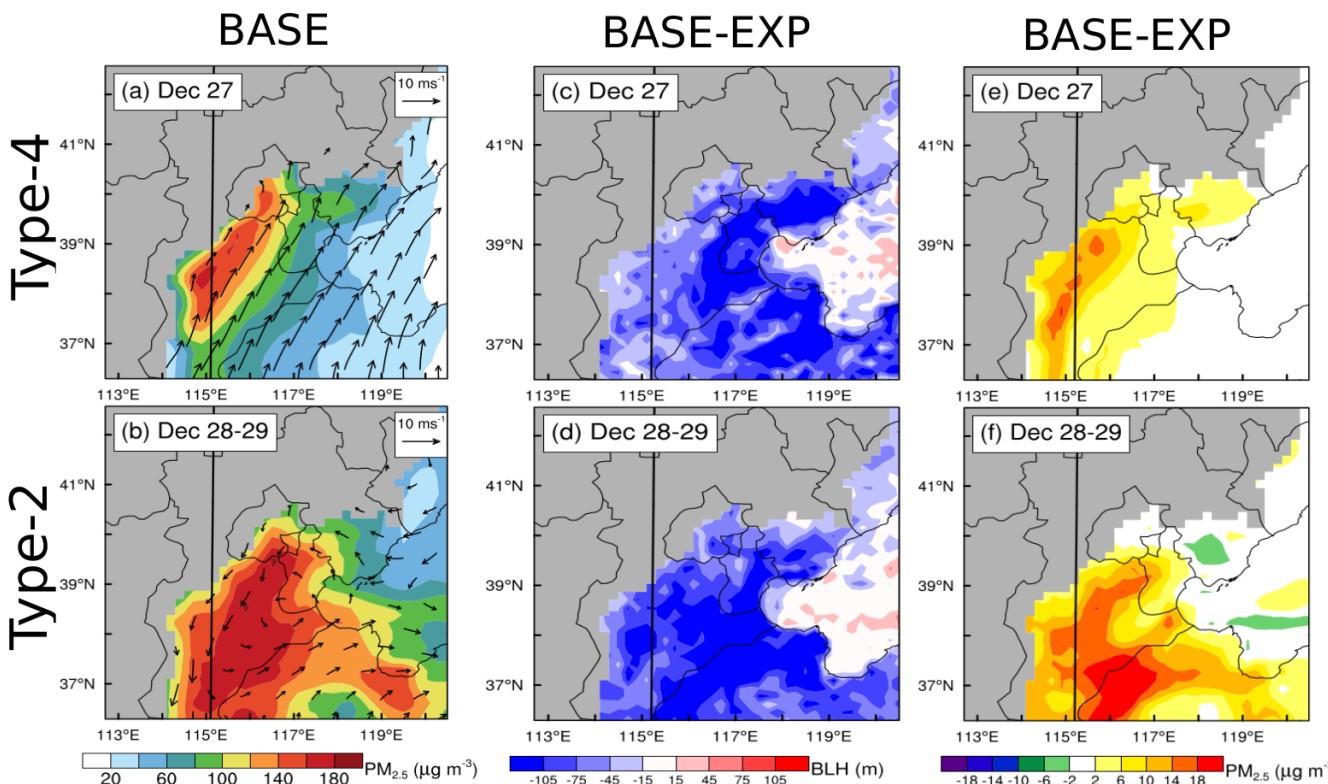

**Figure 9: Spatial distribution of simulated (a, b) near-surface PM$_{2.5}$ concentration and wind, and the perturbations induced by the aerosol radiative effect on (c, d) BLH and (e, f) PM$_{2.5}$ in the plains of BTH during 0900 to 1600 BJT on (top) December 27 and (bottom) December 28-29. The black lines in Fig. 9a indicates the locations of vertical sections shown in Fig. 10.**

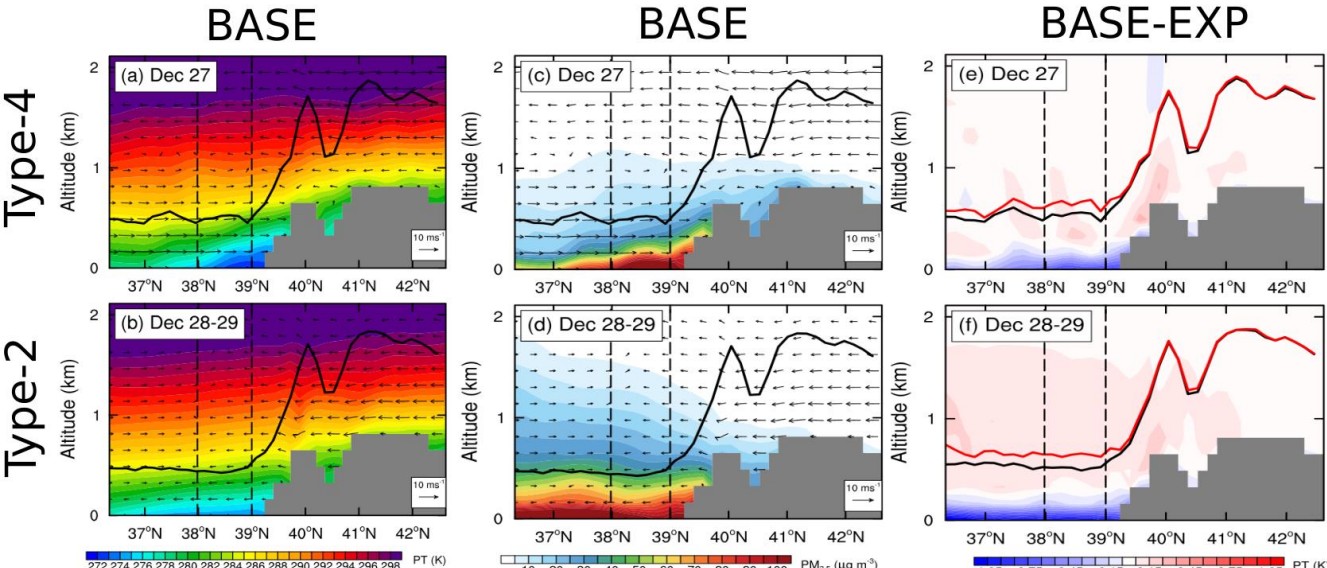

**Figure 10: Vertical cross sections of simulated (a, b) PT, (c, d) PM₂.₅ concentration, and (e, f) the concentration perturbation induced by the aerosol radiative effect during 0900 to 1600 BJT on (top) December 27, and (bottom) December 28-29. The locations of cross section are indicated by the black lines in Fig. 9. In Fig. 10e-f, the BLH of BASE run is denoted by the black lines, and the BLH of EXP run is denoted by the red lines. Note that the vertical velocity is multiplied by a factor of 10 when plotting the wind vectors. The vertical dashed lines indicate the regions to derive the profiles of PT and PM₂.₅ concentration shown in Fig. 11.**

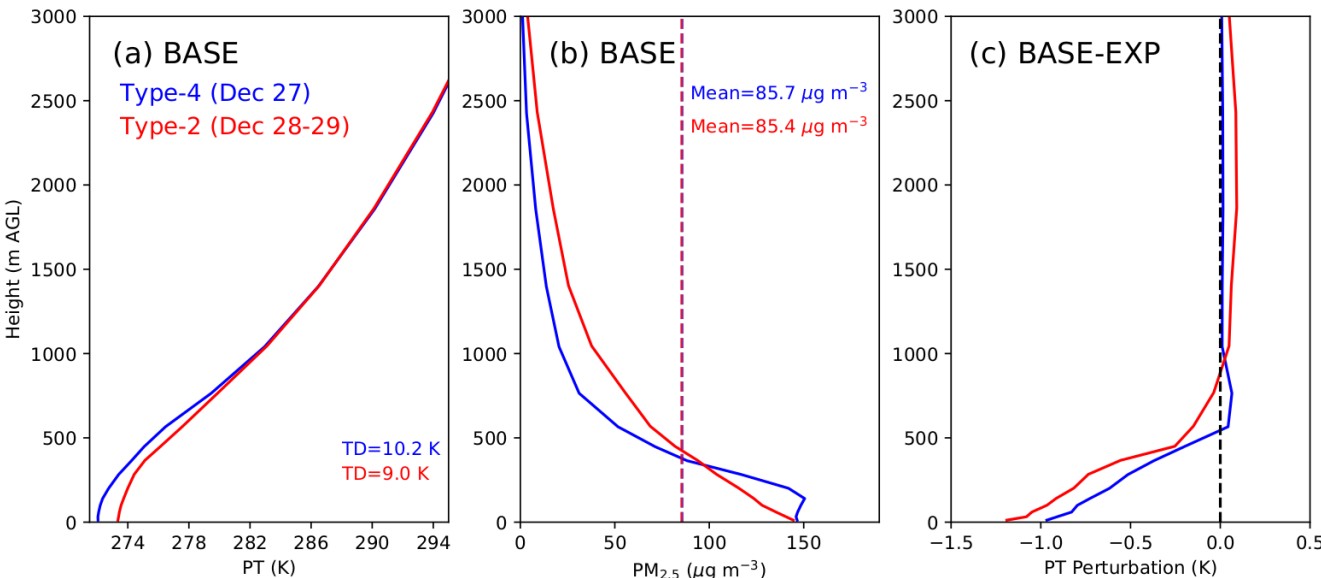

**Figure 11: Average vertical profiles of simulated (a) PT, (b) PM₂.₅ concentration, and (c) PT perturbations induced by the aerosol radiative effect during 0900 to 1600 BJT on December 27 (in blue) and December 28-29 (in red), derived from the simulations along the cross section shown in Fig. 10 between 38 °N and 39 °N. In Fig. 11a, the TD is calculated as the PT difference between 100 m and 1000 m. In Fig. 11b, the dash lines indicate the mean PM₂.₅ concentrations below 3000 m AGL on December 27 (in blue) and December 28-29 (in red).**