# Peer review of "Integrated impacts of synoptic forcing and aerosol radiative effect on boundary layer and pollution in the Beijing-Tianjin-Hebei region, China"

_Atmospheric Chemistry and Physics, 2019_

## Referee Comment (RC1) · Anonymous Referee #1 · 24 Dec 2019

The authors aim to investigate the complicated impacts of synoptic forcing and aerosol radiative effect on boundary layer and pollution in the Beijing-Tianjin-Hebei region of China. The manuscript has well-presented some findings. However, there are still some minor concerns that need to be addressed.

1. Most of all, more deeper analyses are needed for all of the figures. In the current version, the analysis is too simple and rough for the figures. Besides, a mechanism analysis should be performed according to the phenomena. 2. In the Abstract section, Line 10-15, the meaning of "To unravel the complicated impacts of large-scale atmo-
spheric forcing and the local-scale planetary boundary layer (PBL) characteristics on the pollution there" is unclear. Moreover, the title illustrated that the focus of this study is the "impacts of synoptic forcing and aerosol radiative effect on boundary layer and pollution". The Abstract needs more improvements. 3. What is the standard to identify the heavy pollution episodes in Figure 2? 4. The abbreviation should be used in the following illustration after definition. 5. Section 2 should be separated the two parts including "Data" and "model description". 6. In Figure 6, the legend should be given in (a). 7. Figure 7-9, are these results the model simulation or reanalysis results? It should be present clearly in figure title. 8. The black line in Figure 7 is not clear. 9. In Figure 8-9, what is the meaning of gray color shading? In Figure 10a and 10b, what does the white color denote? 10. Page 5, Line 151, the sentence "As the estimated BLH shown in Fig. 5" need to be rephrased. 11. What is the meaning of "region of interest"? Some abbreviation is not needed. For example, ROI,

---

## Referee Comment (RC2) · Anonymous Referee #2 · 31 Dec 2019

This work tried to understand integrated impacts of synoptic forcing and aerosol radiative effect on boundary layer and pollution in the BTH region based on weather typing as well as chemistry-meteorology coupled regional model. I think it is an interesting topic of great importance. By combining observed data together with simulations, the author analyzed the impact of different synoptic patterns and aerosol radiative effect on heavy haze pollution in BTH. The influence of the primary synoptic type and aerosols' feedback are displayed very well separately, while the joint effect of these two processes are not very clear. For example, which synoptic type is more conducive for

the feedback formation and why? Are the differences of pollution level under different synoptic patterns due primarily to regional circulation or intensity of aerosol radiative impact and even more secondary aerosol formation? Overall, more in-depth analysis ought to be provided. Here are some issues that need to be addressed for further improving this work.

Major comments:

This study used T-PCA method to identify main synoptic weather in Section 3.1. I wonder if the sample size is too small to get the representative results. Usually, years of GPH data was utilized for weather classification (Zhang et al.,2016). Another, the domain of used FNL data is not very clear. Did the author just use the FNL data in BTH region as shown in Fig. 3? Can this region well capture the various spatial-scale circulation systems, especially large-scale ones? At last, it seems that Type 4 is more polluted than Type 2 and occurred during 28-31 Dec. in the following case discussed in 3.2, why the synoptic type 2 can be regarded as the representative polluted pattern (Line 133-135)?

One strength of this work is comprehensive observational data. Here, modeled meteorological conditions like air temperature wind speed and RH was validated in detail. However, the modeled air pollution, especially aerosol reproduction, ought to be evaluated since that this work mainly focused on aerosols' impact on meteorology. Thus, the WRF-Chem simulation with/without aerosol radiative effects is suggested to be compared with observed temperature, RH and hourly PM2.5 concentration.

Many previous studies on aerosols' impact on PBL have highlighted the important role of absorbing aerosol (Huang et al.,2018), did it also hold true in these two typical pollution events discussed here? Furthermore, the vertical profile of aerosol, which is highly dependent on synoptic condition, has been proven to play a vital role in aerosols' impacts on PBL development (Wang et al., 2018). It is a very crucial feature related to both synoptic weather and also PBL evolution. Thus, this work could be greatly

improved by drilling down further into the link among synoptic condition, aerosol vertical structure and its impact on PBL, and in turn,air pollution itself.

Minor corrections:

Line 63: In the sentence "... leading to lower the BLH and deteriorate the pollution", "to" is a preposition and should be followed by substantive expressions instead of the root form of a verb.

Line 66: "...meteorological driving for..." should be "...meteorological driving factor for ...".

Line 77: "green triangle" should be in plural form.

Line 85: "... has been widely to ..." should be "...has been widely applied to ...".

Line 88: The same problem as Line 63, "in consideration of" should be followed by substantive expressions not an independent sentence.

Line 116: According to Fig2, it seems that not all the warming of upper air leads to a pollution aggravation (such as the time period at the end of November). Are there any other factors to be mentioned that control the variations of particulate matter in BTH? Besides, the author only gives the variations of potential temperature, while the definition of inversion is more concerned about air temperature. The vertical structure of air temperature may also worth attention.

Line 128: The total occurrence of type 1 and type 2 synoptic pattern is about 70% (Line 123), it's a little confusing why the rate of other synoptic types is no more than 12.5%. Shouldn't they be summed up to 100%?

Line 133: The author may intend to mention "Fig. 4b" instead of "Fig. 3b" since Fig 4 gives the difference of potential temperature.

Line 155: "resulting to" should be corrected to "resulting in".

Figure 1 caption: "green triangle" should be in plurality, i.e. "green triangles".

Section 3.3 is too short to be a subsection

Reference:

Zhang Y, Ding A, Mao H, et al. Impact of synoptic weather patterns and inter-decadal climate variability on air quality in the North China Plain during 1980–2013[J]. Atmospheric environment, 2016, 124: 119-128.

Huang X, Wang Z, Ding A. Impact of Aerosol‐PBL Interaction on Haze Pollution: Multiyear Observational Evidences in North China[J]. Geophysical Research Letters, 2018, 45(16): 8596-8603.

Wang Z, Huang X, Ding A. Dome effect of black carbon and its key influencing factors: a one-dimensional modelling study[J]. Atmospheric Chemistry and Physics, 2018, 18(4): 2821-2834.
* * *

---

## Author Comment (AC2) · 9 Mar 2020

**Reviewer #2**

This work tried to understand integrated impacts of synoptic forcing and aerosol radiative effect on boundary layer and pollution in the BTH region based on weather typing as well as chemistry-meteorology coupled regional model. I think it is an interesting topic of great importance. By combining observed data together with simulations, the author analyzed the impact of different synoptic patterns and aerosol radiative effect on heavy haze pollution in BTH. The influence of the primary synoptic type and aerosols' feedback are displayed very well separately, while the joint effect of these two processes are not very clear. For example, which synoptic type is more conducive for the feedback formation and why? Are the differences of pollution level under different synoptic patterns due primarily to regional circulation or intensity of aerosol radiative impact and even more secondary aerosol formation? Overall, more in-depth analysis ought to be provided. Here are some issues that need to be addressed for further improving this work.

Thanks for reviewing our manuscript and giving constructive suggestions to improve our manuscript! We have carefully revised the manuscript, and tried our best to analyze the joint effect during the aerosol pollution in BTH. Instead of analyzing those two episodes roughly, in the revised manuscript we focused on the pollution episode at the end of December 2017 and tried to unravel the links among the evolution of synoptic conditions, PBL structure, pollutant transport, as well as the aerosol radiative effect. It's found the Type 2 is more conducive for the feedback than Type 4, since more aerosols can be transported to the upper levels (Wang et al., 2018).

In the revised manuscript, the simulations with/without the aerosol radiative effect were compared. Although the aerosol radiative effect can modulate the pollution level, the general variations of aerosol concentration were governed by the evolution of synoptic weather. For the secondary aerosol formation, it is also relevant to the occurrence of certain stagnant synoptic conditions. The variation of the synoptic patterns modulated the ambient pollutants and likely provided the primary driving force for the day-to-day variations in air pollution level (e.g., Chen et al., 2008; Wei et al., 2011; Zhang et al., 2012; Hu et al., 2014; Zhang et al., 2016; Ye et al., 2016; Miao et al., 2017).

**Major comments:**

1. This study used T-PCA method to identify main synoptic weather in Section 3.1. I wonder if the sample size is too small to get the representative results. Usually, years of GPH data was utilized for weather classification (Zhang et al., 2016). Another, the domain of used FNL data is not very clear. Did the author just use the FNL data in BTH region as shown in Fig. 3? Can this region well capture the various spatial-scale circulation systems, especially large-scale ones? At last, it seems that Type 4 is more polluted than Type 2 and occurred during 28-31 Dec. in the following case discussed in 3.2, why the synoptic type 2 can be regarded as the representative polluted pattern (Line 133-135)?

In the synoptic classification, the domain of FNL data used was centered the BTH and covered an area of 106-126 °E in longitude and 29-49 °N in latitude, which was also the domain of WRF-Chem simulation (Fig. 1a). Unlike the previous work of Zhang et al. (2016) that investigated the linkage between East Asian Monsoon, synoptic weather, and air quality in the North China Plain from 1980 to 2013, this study focused on the day-to-day variation of wintertime synoptic condition in BTH during 2017 and 2018. Although both applied weather typing techniques, Zhang et al. (2016) focused on the inter-annual/decadal variability of monsoon, while we focused on the day-to-day evolution of mid-latitude systems in winter. Thus, a relatively smaller domain of interest was used in this study. By examining the simulated GH fields during the selected pollution episode, it's found that the day-to-day evolutions of mid-latitude systems that influence the BTH can be well captured in the domain of interest. In the revised manuscript, the previous work of Zhang et al. (2016) was cited in the Introduction to support the important roles of synoptic weather in air pollution.

To address the concern about data length (8 months, 240 days), we have tested the synoptic classification by extending the data to 24 months (winter months from 2013 to 2018, 721 days). As shown below, similar patterns (Types 1, 2, and 4) were identified. Since we primarily analyzed the PBL structure and pollution levels under those patterns, the sample size would not affect the main findings. The 8-month data were long enough to identify the typical patterns associated with the heavy pollution in winter (Types 2

and 4).

For the representative polluted pattern, we not only considered the pollution level but also the occurrence frequency. Therefore, the Type 2 is regarded as the representative pattern. We agreed with the reviewer that the Type 4 is another important pattern associated with the heavy pollution in BTH, despite of its low occurrence frequency (~5 %). In the revised manuscript, the influences of Type 4 on PBL structure and aerosol transport during the selected episode were analyzed and compared with those of Type 2.

[Figure]

Fig. R1. The 850-hPa geopotential height (GH) fields and wind vectors for the seven classified patterns, identified using the T-PCA method and wintertime reanalysis data from 2013 to 2018. The frequency of each synoptic pattern are also given.

2. One strength of this work is comprehensive observational data. Here, modeled meteorological conditions like air temperature wind speed and RH was validated in detail. However, the modeled air pollution, especially aerosol reproduction, ought to be evaluated since that this work mainly focused on aerosols' impact on meteorology. Thus, the WRF-Chem simulation with/without aerosol radiative effects is suggested to be compared with observed temperature, RH and hourly PM2.5 concentration.

Thanks for you kind suggestion! In the revised manuscript, more observations (e.g., 1000-m air temperature and 200-m relative humidity) were added to characterize the

PBL of different synoptic types (Fig. R2). Among those seven identified synoptic patterns, Types 2 and 4 are associated with heavier pollution level. Both types are characterized by stronger thermal stability (Fig. R2b), warmer upper air (Fig. R2c), and higher near-surface relative humidity (Fig. R2d).

For the validation of WRF-Chem model, the simulations with/without aerosol radiative effect were compared with the observed temperature, relative humidity (RH), and $PM_{2.5}$ concentration in Beijing and Tangshan (Fig. R3). The variations of temperature, RH and $PM_{2.5}$ concentration can be generally well reproduced by the model, although discrepancies existed. Comparing the simulations with aerosol radiative effect (BASE) to those without (EXP), the BASE has slightly higher $PM_{2.5}$ concentrations, higher RHs and lower temperatures, which lead to higher correlation coefficients with the observations.

[Figure]

Fig. R2 (a) Averaged $PM_{2.5}$ concentrations under different synoptic conditions in Beijing, and Tangshan, and associated (b) thermal differences (TD) of PT between 100 m and 1000 m, (c) temperature at 1000 m, and (d) relative humidity at 200 m. The TD equals PT at 1000-m minus PT at 100-m. The size of circle represents the occurrence frequency of each synoptic type. All the meteorological variables shown are derived from the sounding data.

[Figure]

Fig. R3 Time series of observed and simulated (a, d) 2-m temperature (T), (b, e) 2-m relative humidity (RH), and (c, f) $PM_{2.5}$ concentration in (left) Beijing and (right) Tangshan from 26 to 31 December, 2017. The simulations of BASE run are denoted in blue lines, and those of EXP run are denoted in red lines. The correlation coefficients (R) between the observations and simulations are also given for each panel.

3. Many previous studies on aerosols' impact on PBL have highlighted the important role of absorbing aerosol (Huang et al., 2018), did it also hold true in these two typical pollution events discussed here? Furthermore, the vertical profile of aerosol, which is highly dependent on synoptic condition, has been proven to play a vital role in aerosols' impacts on PBL development (Wang et al., 2018). It is a very crucial feature related to both synoptic weather and also PBL evolution. Thus, this work could be greatly improved by drilling down further into the link among synoptic condition, aerosol vertical structure and its impact on PBL, and in turn air pollution itself.

Yes, the absorbing aerosol played an important role in the PBL structure and aerosol pollution during the selected episode, due to the considerable amount of light-absorbing components in the aerosols of northern China (Ding et al., 2016). In the revised

manuscript, we compared our results with the literatures recommended and further discussed the critical roles of absorbing aerosol.

Besides, the links among synoptic condition, aerosol vertical structure and PBL at the end of December 2017 were analyzed. The BTH was influenced by the synoptic Type 4 on December 27, which turned into the synoptic Type 2 on December 28-29 (Figs. R4 and R5). Both Types led to the warming of upper air and the suppression of PBL (Figs. R4, R5, and R6), which favor the accumulation of pollutants (Figs. R4, R5a-b and R6c-d). These results indicated that the evolution of synoptic pattern provided the primary driving force for the day-to-day variations of aerosol concentrations during the episode.

Besides, influencing by the Types 4 and 2, prominent perturbations induced by the aerosol radiative effect on the BLH and $PM_{2.5}$ concentrations could be found (Fig. R5c-f). It seems that Type 2 is more conducive for the aerosol radiative feedback (Figs. R6 and R7) than Type 4. Comparing with Type 2, the Type 4 induced a stronger thermal stability in the lower troposphere (Figs. R6a-b and R7a), and trapped more aerosols below 400 m AGL (Figs. R6c-d and R7b). By contrast, more aerosols were transported to upper levels under Type 2 (Figs. R6c-d and R7b), which can enhance the aerosol-PBL radiative feedback (Figs. R6e-f and R7c) due to the relatively stronger solar radiation and weaker turbulence at the levels close to the PBL top (Wang et al., 2018).

[Figure]

Fig. R4 Simulated 850-hPa (a-c) GH fields and (d-f) PT fields on December 27, 28-29 and 30-31, overlaid with the wind vectors. The regions with terrains higher than the 850-hPa level are marked by the grey shadings.

[Figure]

Fig. R5 Spatial patterns of (a-b) simulated near-surface PM$_{2.5}$ concentration, and the perturbations induced by the aerosol radiative effect on (c-d) BLH and (e-f) PM$_{2.5}$ in the plans of BTH during 0900 to 1600 BJT on (top) December 27 and (bottom) December 28-29, 2017. The mountains are marked by the grey shadings. The black line in Fig. R5a indicates the locations of vertical sections shown in Fig. R6.

[Figure]

Fig. R6. Vertical cross sections of simulated (a-b) PT, (c-d) PM$_{2.5}$ concentration, and (e-f) the PM$_{2.5}$ perturbations induced by the aerosol radiative effect during 0900 to 1600 BJT on (top) December 27, and (bottom) December 28-29. The locations of cross section (115.2 °E) are indicated by the black lines in Fig. R5. In Fig. R6e-f, the BLH of BASE run is denoted by the black solid lines, and the BLH of EXP run is denoted by the red lines. Note that the vertical velocity is multiplied by a factor of 10 when plotting the wind vectors. The two dashed lines indicated the regions to derive the vertical profiles of PT and PM$_{2.5}$ concentration shown in Fig. R7.

[Figure]

Fig. R7 Averaged profiles of simulated (a) PT, (b) PM$_{2.5}$ concentration and (c) the PT perturbations induced by the aerosol radiative effect during 0900 to 1600 BJT on December 27 (in blue) and December 28-29 (in red). The profiles shown were derived from the simulations along the cross sections shown in Fig. R6 between 37.5 °N and 39 °N.

**Minor corrections:**

Line 63: In the sentence "…leading to lower the BLH and deteriorate the pollution", "to" is a preposition and should be followed by substantive expressions instead of the root form of a verb.

Thanks, the sentence was revised as: "…the massive aerosols can intensify the PBL stability through scattering the solar radiations, which can lower the BLH and deteriorate the pollution".

Line 66: "…meteorological driving for" should be "…meteorological driving factor for ".

Thanks, the sentence was revised as suggested.

Line 77: "green triangle" should be in plural form.

Revised as suggested.

Line 85: "… has been widely to …" should be "… has been widely applied to …".

Revised as suggested.

Line 88: The same problem as Line 63, "in consideration of" should be followed by substantive expressions not an independent sentence.

The sentence was revised as: "Considering that the heavy $PM_{2.5}$ pollution events primarily occurred during winter …"

Line 116: According to Fig. 2, it seems that not all the warming of upper air leads to a pollution aggravation (such as the time period at the end of November). Are there any other factors to be mentioned that control the variations of particulate matter in BTH? Besides, the author only gives the variations of potential temperature, while the definition of inversion is more concerned about air temperature. The vertical structure of air temperature may also worth attention.

To understand the relationships between the warming of upper air and the variations of

PM$_{2.5}$ concentration, we compared the daily variations of PM$_{2.5}$ concentration ($\Delta$PM$_{2.5}$) and those of 1300-m PT ($\Delta$PT) in Beijing in November and December 2017. As shown in Fig. R8, there were 32 days associated with the warming of upper air over Beijing, and all these days were corresponding to the increase of PM$_{2.5}$ concentration. Similar relationship also can be found in Tangshan, which implies the critical role of thermal stability in the aerosol pollution of BTH during winter.

Thanks for your kind suggestion, in the revised manuscript the vertical structure of air temperature (Fig. R2) were also examined to further understand the relationships between PBL thermal structure and aerosol pollution.

We agreed with the reviewer that there are other factors that influence the aerosol pollution level in BTH, in addition to the local PBL thermal structure, such as the transport of pollutants. Thus, in the revised manuscript, the transport of pollutants associated with the synoptic Type 4 during the selected episode (Fig. R5a) were analyzed and discussed.

[Figure]

Fig. R8 (a) Time series of observed PM$_{2.5}$ concentration from 1 November to 31 December in 2017 in Beijing, and (b) the 24-hr variations of potential temperature ($\Delta$PT) derived from the sounding data at 2000 BJT. In Fig. R8c, the daily variations of PM$_{2.5}$ concentration ($\Delta$PM$_{2.5}$) are denoted by the color bars (positive values in red, negative

values in blue), and those of 24-hr ΔPT at 1300 m are shown by the color pluses (positive values in red, negative values in blue).

Line 128: The total occurrence of type 1 and type 2 synoptic pattern is about 70% (Line 123), it's a little confusing why the rate of other synoptic types is no more than 12.5%. Shouldn't they be summed up to 100%?

Yes, all the types were summed up to 100%. Sorry for this ambiguous sentence, which was revised as: "Except for these two dominant types, the occurrence rate of other five synoptic types is 30.4% in total."

Line 133: The author may intend to mention "Fig. 4b" instead of "Fig. 3b" since Fig 4 gives the difference of potential temperature.

Sorry for this typo. It was corrected in the revised manuscript.

Line 155: "resulting to" should be corrected to "resulting in".

Revised as suggested.

Figure 1 caption: "green triangle" should be in plurality, i.e. "green triangles".

Revised as suggested.

Section 3.3 is too short to be a subsection

The Section 3.3 was merged into the previous section as suggested.

Reference:

Chan C et al., Air pollution in mega cities in China, Atmos. Environ., 2008, 42, 1–42, doi:10.1016/j.atmosenv.2007.09.003, 2008.

Ding et al., Enhanced haze pollution by black carbon in megacities in China, Geophys. Res. Lett., 2016, 43, doi:10.1002/2016GL067745

Hu X et al., Impact of the Loess Plateau on the atmospheric boundary layer structure and air quality in the North China Plain? A case study, Sci. Total Environ., 2014, 499,

228–237, doi:10.1016/j.scitotenv.2014.08.053

Huang X et al., Impact of Aerosol-PBL Interaction on Haze Pollution: Multiyear Observational Evidences in North China. Geophysical Research Letters, 2018, 45(16): 8596-8603.

Miao Y et al., Classification of summertime synoptic patterns in Beijing and their associations with boundary layer structure affecting aerosol pollution, Atmos. Chem. Phys., 2017, 17(4), 3097–3110

Wang Z et al., Dome effect of black carbon and its key influencing factors: a one-dimensional modelling study. Atmospheric Chemistry and Physics, 2018, 18(4): 2821-2834.

Wei P et al., Impact of boundary-layer anticyclonic weather system on regional air quality, Atmos. Environ., 2011, 45, 2453–2463, doi:10.1016/j.atmosenv.2011.01.045.

Ye X et al., Study on the synoptic flow patterns and boundary layer process of the severe haze events over the North China Plain in January 2013, Atmos. Environ., 2016, 124, 129–145.

Zhang J et al., The impact of circulation patterns on regional transport pathways and air quality over Beijing and its surroundings, Atmos. Chem. Phys., 2012, 12, 5031–5053, doi:10.5194/acp-12-5031-2012.

Zhang Y et al., Impact of synoptic weather patterns and inter-decadal climate variability on air quality in the North China Plain during 1980–2013. Atmospheric environment, 2016, 124: 119-128.

---

## Author Response (AR1)

**Reviewer #1**

The authors aim to investigate the complicated impacts of synoptic forcing and aerosol radiative effect on boundary layer and pollution in the Beijing-Tianjin-Hebei region of China. The manuscript has well-presented some findings. However, there are still some minor concerns that need to be addressed.

Thanks for taking time to review our manuscript and offer helpful suggestions! We carefully revised the manuscript, please see the response below.

1. Most of all, more deeper analyses are needed for all of the figures. In the current version, the analysis is too simple and rough for the figures. Besides, a mechanism analysis should be performed according to the phenomena.

More analyses and discussions were added for most figures as suggested. For the associations between synoptic pattern and PBL structure, more meteorological parameters (e.g., temperature and relatively humidity) were compared and analyzed. In addition to the synoptic Type 2, the impacts of Type 4 on PBL and aerosol pollution were also elucidated based on the long-term soundings and PM$_{2.5}$ measurements.

Besides, instead analyzing those two episodes roughly, in the revised manuscript we focused on the pollution episode at the end of December 2017 and tried our best to unravel the links among the evolution of synoptic conditions (i.e., Type 4 on December 27, Type 2 on December 28-29, Type 1 on December 30-31), PBL structure, aerosol vertical distribution, as well as the aerosol radiative effect. The possible mechanisms related to the development of PBL were carefully given based on the abovementioned observational and simulated analysis.

2. In the Abstract section, Line 10-15, the meaning of "To unravel the complicated impacts of large-scale atmospheric forcing and the local-scale planetary boundary layer (PBL) characteristics on the pollution there" is unclear. Moreover, the title illustrated that the focus of this study is the "impacts of synoptic forcing and aerosol radiative effect on boundary layer and pollution". The Abstract needs more improvements.

The sentence mentioned were rewritten as:

*"The heavy aerosol pollutions frequently occur in winter, closely in relation to the planetary boundary layer (PBL) meteorology. To unravel the physical processes that influence the PBL structure and aerosol pollution in BTH, this study combined long-term observational data analyses, synoptic pattern classification, and meteorology-chemistry coupled simulations."*

Since more analyses were added, most of the abstract were re-written and highlighted the integrated impacts of synoptic pattern and aerosol radiative effect.

3. What is the standard to identify the heavy pollution episodes in Figure 2?

The heavy pollution episode was identified when the maximum daily $PM_{2.5}$ concentration is greater than 100 μg m$^{-3}$ in both Beijing and Tangshan. The relevant information was added in the figure title.

[Figure]

Fig. 2. Time series of observed $PM_{2.5}$ concentration from 1 November to 31 December in 2017 in (a) Beijing and Tangshan, and (b, c) vertical structure of potential temperature (PT) derived from the sounding data at 2000 BJT. Four heavy pollution episodes with maximum daily $PM_{2.5}$ concentration greater than 100 μg m$^{-3}$ in both

Beijing and Tangshan are marked by the grey shadings in Fig. 2a.

4. The abbreviation should be used in the following illustration after definition.

Thanks for your kind suggestion. The definitions of abbreviation were added in revised manuscript, and some abbreviations were removed (e.g., ROI).

5. Section 2 should be separated the two parts including "Data" and "model description".

The Section 2 was separated as suggested.

6. In Figure 6, the legend should be given in (a).

Revised as suggested.

7. Figure 7-9, are these results the model simulation or reanalysis results? It should be present clearly in figure title.

All these figures presented the model simulations. The information was clearly stated in the figure titles as suggested.

8. The black line in Figure 7 is not clear.

In the revised manuscript, we reorganized the manuscript and removed the Figure 7.

9. In Figure 8-9, what is the meaning of gray color shading? In Figure 10a and 10b, what does the white color denote?

The grey color shadings in Figs. 8-9 denote the mountains. In Fig. 10a and 10b, the white color shadings also denotes the mountains. In the revised manuscript, all the mountains were denoted by the grey shadings and clearly stated in the figure titles.

10. Page 5, Line 151, the sentence "As the estimated BLH shown in Fig. 5" need to be rephrased.

In the revised manuscript, the sentence was revised as: "Fig. 5 shows the time series of simulated BLH in Beijing and Tangshan."

11. What is the meaning of "region of interest"? Some abbreviation is not needed. For example, ROI.

The "region of interest" is the region we primarily focused on. To be clear, in the revised manuscript, the abbreviation was removed as suggested.

**Reviewer #2**

This work tried to understand integrated impacts of synoptic forcing and aerosol radiative effect on boundary layer and pollution in the BTH region based on weather typing as well as chemistry-meteorology coupled regional model. I think it is an interesting topic of great importance. By combining observed data together with simulations, the author analyzed the impact of different synoptic patterns and aerosol radiative effect on heavy haze pollution in BTH. The influence of the primary synoptic type and aerosols' feedback are displayed very well separately, while the joint effect of these two processes are not very clear. For example, which synoptic type is more conducive for the feedback formation and why? Are the differences of pollution level under different synoptic patterns due primarily to regional circulation or intensity of aerosol radiative impact and even more secondary aerosol formation? Overall, more in-depth analysis ought to be provided. Here are some issues that need to be addressed for further improving this work.

Thanks for reviewing our manuscript and giving constructive suggestions to improve our manuscript! We have carefully revised the manuscript, and tried our best to analyze the joint effect during the aerosol pollution in BTH. Instead of analyzing those two episodes roughly, in the revised manuscript we focused on the pollution episode at the end of December 2017 and tried to unravel the links among the evolution of synoptic conditions, PBL structure, aerosol vertical distribution, as well as the aerosol radiative effect. It's found the Type 2 is more conducive for the feedback than Type 4, since more aerosols can be transported to the upper levels (Wang et al., 2018).

In the revised manuscript, the simulations with/without the aerosol radiative effect were compared. Although the aerosol radiative effect can modulate the pollution level, the general variations of aerosol concentration were governed by the evolution of synoptic weather. For the secondary aerosol formation, it is also relevant to the occurrence of certain stagnant synoptic conditions. The variation of the synoptic patterns modulated the ambient pollutants and likely provided the primary driving force for the day-to-day variations in air pollution level (e.g., Chen et al., 2008; Wei et al., 2011; Zhang et al., 2012; Hu et al., 2014; Zhang et al., 2016; Ye et al., 2016; Miao et al., 2017).

**Major comments:**

1. This study used T-PCA method to identify main synoptic weather in Section 3.1. I wonder if the sample size is too small to get the representative results. Usually, years of GPH data was utilized for weather classification (Zhang et al., 2016). Another, the domain of used FNL data is not very clear. Did the author just use the FNL data in BTH region as shown in Fig. 3? Can this region well capture the various spatial-scale circulation systems, especially large-scale ones? At last, it seems that Type 4 is more polluted than Type 2 and occurred during 28-31 Dec. in the following case discussed in 3.2, why the synoptic type 2 can be regarded as the representative polluted pattern (Line 133-135)?

In the synoptic classification, the domain of FNL data used was centered the BTH and covered an area of 106-126 °E in longitude and 29-49 °N in latitude, which was also the domain of WRF-Chem simulation (Fig. 1a). Unlike the previous work of Zhang et al. (2016) that investigated the linkage between East Asian Monsoon, synoptic weather, and air quality in the North China Plain from 1980 to 2013, this study focused on the day-to-day variation of wintertime synoptic condition in BTH during 2017 and 2018. Although both applied weather typing techniques, Zhang et al. (2016) focused on the inter-annual/decadal variability of monsoon, while we focused on the day-to-day evolution of mid-latitude systems in winter. Thus, a relatively smaller domain of interest was used in this study. By examining the simulated GH fields during the selected pollution episode, it's found that the day-to-day evolutions of mid-latitude systems that influence the BTH can be well captured in the domain of interest. In the revised manuscript, the previous work of Zhang et al. (2016) was cited in the Introduction to support the important roles of synoptic weather in air pollution.

To address the concern about data length (8 months, 240 days), we have tested the synoptic classification by extending the data to 24 months (winter months from 2013 to 2018, 721 days). As shown below, similar patterns (Types 1, 2, and 4) were identified. Since we primarily analyzed the PBL structure and pollution levels under these patterns, the sample size cannot affect the main findings in this study. The 8-month data were long enough to identify the typical patterns associated with the heavy pollution in winter

(Types 2 and 4).

For the representative polluted pattern, we not only considered the pollution level but also the occurrence frequency. Therefore, the Type 2 is regarded as the representative pattern. We agreed with the reviewer that the Type 4 is another important pattern associated with the heavy pollution in BTH, despite of its low occurrence frequency (~5 %). In the revised manuscript, the influences of Type 4 on PBL structure and aerosol transport during the selected episode were analyzed and compared with those of Type 2.

[Figure]

Fig. R1. The 850-hPa geopotential height (GH) fields and wind vectors for the seven classified patterns, identified using the T-PCA method and wintertime reanalysis data from 2013 to 2018. The frequency of each synoptic pattern is also given.

2. One strength of this work is comprehensive observational data. Here, modeled meteorological conditions like air temperature wind speed and RH was validated in detail. However, the modeled air pollution, especially aerosol reproduction, ought to be evaluated since that this work mainly focused on aerosols' impact on meteorology. Thus, the WRF-Chem simulation with/without aerosol radiative effects is suggested to be compared with observed temperature, RH and hourly $PM_{2.5}$ concentration.

Thanks for you kind suggestion! In the revised manuscript, more observations (e.g., 1000-m air temperature and 200-m relative humidity) were added to characterize the

PBL of different synoptic types (Fig. 4). Among those seven identified synoptic patterns, Types 2 and 4 are associated with heavier pollution level. Both types are characterized by stronger thermal stability (Fig. 4b), warmer upper air (Fig. 4c), and higher near-surface relative humidity (Fig. 4d).

For the validation of WRF-Chem model, the simulations with/without aerosol radiative effect were compared with the observed temperature, relative humidity (RH), and $PM_{2.5}$ concentration in Beijing and Tangshan (Fig. 6). The variations of temperature, RH and $PM_{2.5}$ concentration can be generally well reproduced by the model, although discrepancies existed. Comparing the simulations with aerosol radiative effect (BASE) to those without (EXP), the BASE has slightly higher $PM_{2.5}$ concentrations, higher RHs and lower temperatures, which lead to higher correlation coefficients with observations. In the revised manuscript, the relevant analyses were added in the Sections 3.1 and 3.2.

[Figure]

Fig. 4. (a) Average $PM_{2.5}$ concentrations under different synoptic conditions in Beijing and Tangshan, and associated (b) thermal differences (TD) of PT between 100 m and 1000 m, and (c) temperature at 1000 m, and (d) relative humidity (RH) at 200 m. The TD equals PT at 1000 m minus PT at 100 m. The size of circle represents the occurrence frequency of each synoptic type. All the meteorological variables shown are derived from the radiosonde data.

[Figure]

Fig. 6. Time series of observed and simulated (a, b) 2 m temperature, (c, d) 2 m RH, and (e, f) PM2.5 concentration in (left) Beijing and (right) Tangshan from 26 to 31 December 2017. The simulations of BASE run are denoted in blue lines, and those of EXP run are denoted in red lines. The correlation coefficients (R) between the observations and simulations are also given for each panel.

3. Many previous studies on aerosols' impact on PBL have highlighted the important role of absorbing aerosol (Huang et al., 2018), did it also hold true in these two typical pollution events discussed here? Furthermore, the vertical profile of aerosol, which is highly dependent on synoptic condition, has been proven to play a vital role in aerosols' impacts on PBL development (Wang et al., 2018). It is a very crucial feature related to both synoptic weather and also PBL evolution. Thus, this work could be greatly improved by drilling down further into the link among synoptic condition, aerosol vertical structure and its impact on PBL, and in turn air pollution itself.

Yes, the absorbing aerosol played an important role in the PBL structure and aerosol pollution during the selected episode, due to the considerable amount of light-absorbing components in the aerosols of northern China (Ding et al., 2016). In the revised

manuscript, the critical roles of absorbing aerosols were discussed and the recommended literatures were properly cited.

Besides, the links among synoptic condition, aerosol vertical structure and PBL at the end of December 2017 were analyzed. The BTH was influenced by the synoptic Type 4 on December 27, which turned into the synoptic Type 2 on December 28-29 (Figs. 7 and 9). Both Types can lead to the warming of upper air and the suppression of PBL (Figs. 7), favoring the accumulation of pollutants (Figs. 7, 9a-b and 10c-d). These results indicated that the evolution of synoptic pattern provided the primary driving force for the day-to-day variations of aerosol concentrations during the studied episode. Also, influencing by the Types 4 and 2, prominent perturbations induced by the aerosol radiative effect on the BLH and $PM_{2.5}$ concentrations could be found (Fig. 9c-f). It seems that Type 2 is more conducive for the aerosol radiative feedback (Figs. 10 and 11) than Type 4. Comparing with Type 2, the Type 4 induced a stronger thermal stability in the lower troposphere (Figs. 10a-b and 11a), and trapped more aerosols below 400 m AGL (Figs. 10c-d and 11b). By contrast, more aerosols were transported to upper levels under Type 2 (Figs. 10c-d and 11b), which can enhance the aerosol-PBL radiative feedback (Figs. 10e-f and 11c) due to the stronger solar radiation and weaker turbulence at the upper levels close to the PBL top (Wang et al., 2018).

In the revised manuscript, the relevant analyses and discussions were added in the Section 3.2.

[Figure]

Fig. 7. Simulated 850-hPa (a-c) GH and (d-f) PT fields on December 27, 28-29 and 30-31, overlaid with the wind vectors. The regions with terrains higher than the 850-hPa level are marked by the grey shadings.

[Figure]

Fig. 9. Spatial distribution of simulated (a, b) near-surface PM$_{2.5}$ concentration and wind, and the perturbations induced by the aerosol radiative effect on (c, d) BLH and (e, f) PM$_{2.5}$ in the plains of BTH during 0900 to 1600 BJT on (top) December 27 and (bottom) December 28-29. The black lines in Fig. 9a indicates the locations of vertical sections shown in Fig. 10.

[Figure]

Fig. 10. Vertical cross sections of simulated (a, b) PT, (c, d) PM$_{2.5}$ concentration, and (e, f) the concentration perturbation induced by the aerosol radiative effect during 0900 to 1600 BJT on (top) December 27, and (bottom) December 28-29. The locations of cross section are indicated by the black lines in Fig. 9. In Fig. 10e-f, the BLH of BASE run is denoted by the black lines, and the BLH of EXP run is denoted by the red lines. Note that the vertical velocity is multiplied by a factor of 10 when plotting the wind vectors. The vertical dashed lines indicate the regions to derive the profiles of PT and PM$_{2.5}$ concentration shown in Fig. 11.

[Figure]

Fig. 11. Average vertical profiles of simulated (a) PT, (b) PM$_{2.5}$ concentration, and (c) PT perturbations induced by the aerosol radiative effect during 0900 to 1600 BJT on December 27 (in blue) and December 28-29 (in red), derived from the simulations along the cross section shown in Fig. 10 between 38 °N and 39 °N. In Fig. 11a, the TD is calculated as the PT difference between 100 m and 1000 m. In Fig. 11b, the dash lines indicate the mean PM$_{2.5}$ concentrations below 3000 m AGL on December 27 (in blue) and December 28-29 (in red).

**Minor corrections:**

Line 63: In the sentence "…leading to lower the BLH and deteriorate the pollution", "to" is a preposition and should be followed by substantive expressions instead of the root form of a verb.

Thanks, the sentence was revised as: "…the massive aerosols can intensify the PBL stability through scattering the solar radiations, which can lower the BLH and deteriorate the pollution".

Line 66: "…meteorological driving for" should be "…meteorological driving factor for ".

Thanks, the sentence was revised as suggested.

Line 77: "green triangle" should be in plural form.

Revised as suggested.

Line 85: "… has been widely to …" should be "… has been widely applied to …".

Revised as suggested.

Line 88: The same problem as Line 63, "in consideration of" should be followed by substantive expressions not an independent sentence.

The sentence was revised as: "Considering that the heavy PM$_{2.5}$ pollution events primarily occurred during winter …"

Line 116: According to Fig. 2, it seems that not all the warming of upper air leads to a pollution aggravation (such as the time period at the end of November). Are there any other factors to be mentioned that control the variations of particulate matter in BTH? Besides, the author only gives the variations of potential temperature, while the definition of inversion is more concerned about air temperature. The vertical structure of air temperature may also worth attention.

To understand the relationships between the warming of upper air and the variations of

PM$_{2.5}$ concentration, we compared the daily variations of PM$_{2.5}$ concentration ($\Delta$PM$_{2.5}$) and those of 1300-m PT ($\Delta$PT) in Beijing in November and December 2017. As shown in Fig. R2, there were 32 days associated with the warming of upper air over Beijing, and all these days were corresponding to the increases of PM$_{2.5}$ concentration. Similar relationship also can be found in Tangshan, which implies the critical role of thermal stability in the aerosol pollution of BTH during winter.

Thanks for your kind suggestion, in the revised manuscript the vertical structure of air temperature (Fig. 4) were also examined to further understand the relationships between PBL thermal structure and aerosol pollution.

We agreed with the reviewer that in addition to the local PBL thermal structure, there are other factors that influence the aerosol pollution level in BTH, such as the transport of pollutants. Thus, in the revised manuscript, the transport of pollutants associated with the synoptic Type 4 during the selected episode were analyzed and discussed in the Sections 3.1 and 3.2.

[Figure]

Fig. R2 (a) Time series of observed PM$_{2.5}$ concentration from 1 November to 31 December in 2017 in Beijing, and (b) the 24-hr variations of potential temperature ($\Delta$PT) derived from the sounding data at 2000 BJT. In Fig. R2c, the daily variations of PM$_{2.5}$ concentration ($\Delta$PM$_{2.5}$) are denoted by the color bars (positive values in red, negative values in blue), and those of 24-hr $\Delta$PT at 1300 m are shown by the color pluses (positive values in red, negative values in blue).

Line 128: The total occurrence of type 1 and type 2 synoptic pattern is about 70% (Line 123), it's a little confusing why the rate of other synoptic types is no more than 12.5%. Shouldn't they be summed up to 100%?

Yes, all the types were summed up to 100%. Sorry for this ambiguous sentence, which was revised as: "Except for these two dominant types, the occurrence rate of other five synoptic types is 30.4% in total."

Line 133: The author may intend to mention "Fig. 4b" instead of "Fig. 3b" since Fig 4 gives the difference of potential temperature.

Sorry for this typo. It was corrected in the revised manuscript.

Line 155: "resulting to" should be corrected to "resulting in".

Revised as suggested.

Figure 1 caption: "green triangle" should be in plurality, i.e. "green triangles".

Revised as suggested.

Section 3.3 is too short to be a subsection

The Section 3.3 was merged into the previous section as suggested.

[revised manuscript text omitted]